# Independent transcriptional patterns reveal biological processes associated with disease-free survival in early colorectal cancer
Daan G. Knapen [1], Sara Hone Lopez [1], Derk Jan A. de Groot[1], Jacco-Juri de Haan [1], Elisabeth G. E. de Vries [1], Rodrigo Dienstmann[2], Steven de Jong [1], Arkajyoti Bhattacharya [1] & Rudolf S. N. Fehrmann [1] ✉

## Abstract

**Background** Bulk transcriptional profiles of early colorectal cancer (CRC) can fail to detect biological processes associated with disease-free survival (DFS) if the transcriptional patterns are subtle and/or obscured by other processes' patterns. Consensus-independent component analysis (c-ICA) can dissect such transcriptomes into statistically independent transcriptional components (TCs), capturing both pronounced and subtle biological processes.

**Methods** In this study we (1) integrated transcriptomes ($n = 4228$) from multiple early CRC studies, (2) performed c-ICA to define the TC landscape within this integrated data set, 3) determined the biological processes captured by these TCs, (4) performed Cox regression to identify DFS-associated TCs, (5) performed random survival forest (RSF) analyses with activity of DFS-associated TCs as classifiers to identify subgroups of patients, and 6) performed a sensitivity analysis to determine the robustness of our results

**Results** We identify 191 TCs, 43 of which are associated with DFS, revealing transcriptional diversity among DFS-associated biological processes. A prominent example is the epithelial-mesenchymal transition (EMT), for which we identify an association with nine independent DFS-associated TCs, each with coordinated upregulation or downregulation of various sets of genes.

**Conclusions** This finding indicates that early CRC may have nine distinct routes to achieve EMT, each requiring a specific peri-operative treatment strategy. Finally, we stratify patients into DFS patient subgroups with distinct transcriptional patterns associated with stage 2 and stage 3 CRC.

## Plain language summary

While treatments for patients with colorectal cancer have improved, many patients (around 30-50%) have cancers that will eventually relapse and these patients will die due to their disease. Researchers have been studying the genes involved in colorectal cancer to help us understand why some cancers might relapse. However, current methods to do this may miss subtle or hidden patterns in the gene activity related to cancer relapse. To deal with this, we used a special method called consensus-independent component analysis (c-ICA) to dig more deeply into the activity of genes. This helped us to uncover some potential biological processes underpinning colorectal cancer relapse, which ultimately could help researchers to identify better treatments for patients with colorectal cancer.

Although colorectal cancer (CRC) is currently the second-leading cause of cancer-related death worldwide[1], CRC-related mortality has decreased due to early detection followed by curative surgical resection[2]. The most recent advance in peri-operative systemic therapy was in 2004 when oxaliplatin was added to leucovorin-modulated 5-fluorouracil[3]. Despite improvements in the detection and treatment of CRC, however, approximately 30-50% of all patients treated for early CRC relapse and die due to the disease[3]. This alarming statistic underscores the need to gain new insights into the complex biology underlying early CRC and develop more effective treatment strategies.

Previous studies involving the transcriptional profiling of CRC have greatly increased our understanding of the biological processes associated

[1]Department of Medical Oncology, University Medical Center Groningen, University of Groningen, Groningen, the Netherlands. [2]Oncology Data Science (ODysSey) Group, Vall d'Hebron Institute of Oncology, Universitat Autónoma de Barcelona, Barcelona, Spain. ✉e-mail: r.s.n.fehrmann@umcg.nl

with disease-free survival (DFS) and have revealed expression-based molecular subtypes[4–10]. The International CRC Subtyping Consortium has also identified consensus molecular subtypes (CMSs)[11]. In these studies, bulk transcriptional profiling was performed using CRC samples containing both tumor cells and the tumor microenvironment, thus reflecting the average transcriptional patterns of the combination of all biological processes present in the tumors. Thus, if related transcriptional patterns are subtle and/or obscured by other process-related patterns, this approach may fail to detect the biological processes specifically relevant to DFS[12].

To overcome this issue, consensus-independent component analysis (c-ICA) can capture the transcriptional patterns of both robust and subtle biological processes by dissecting the tumor bulk transcriptomes into statistically independent components called transcriptional components (TCs)[13–15]. The activity of these TCs can then be determined in a bulk transcriptional profile.

In the current study, we use c-ICA to identify TCs in early CRCs and gain new insights into the complex biological processes associated with DFS.

## Methods

Detailed information regarding the methods is provided in the Supplementary Materials. An overview of our data analysis approach is presented in Fig. 1.

### Data acquisition

Publicly available raw microarray expression profiles were obtained from the Gene Expression Omnibus (GEO)[16]. Data acquisition was restricted to the Affymetrix HG-U133 Plus 2.0 platform (GEO accession identifier: GPL570). Profiles were selected if they were generated using tissue samples of various healthy and colorectal conditions in the large intestine (Fig. 1A). The CRC samples were all obtained from patients with early CRC. Any duplicate samples were identified and excluded from our analysis. The raw data files were pre-processed and normalized using the robust multiarray average algorithm[17]. Quality control was performed using principal component analysis as previously described[12]. All available clinicopathological data were collected and manually curated for the CRC samples (Fig. 1B); here, we define this data set as the primary data set.

The pre-processed and normalized expression profiles were used to infer the tumor microenvironment composition using the Microenvironment Cell Population (MCP) counter, version 1.1, available as an R package (Fig. 1B)[18]. The MCP counter estimates the absolute abundance of eight immune and two stromal cell populations on bulk gene expression profiles. We utilized the random forest CMS classifier was used to assign CMS subtypes to the CRCs (Fig. 1B)[11].

### Consensus-independent component analysis

c-ICA was subsequently used to dissect the bulk transcriptional profiles into statistically independent TCs, as described previously and in the Supplementary Methods (Fig. 1C, D)[19]. In short, in the dataset containing mRNA expression profiles of p genes from n samples, the output of an ICA includes two matrices: (i) an independent component matrix with dimensions $i \times p$, where each weight within the independent components represents both the direction and magnitude of its effect on the expression levels of each gene, and (ii) a mixing matrix with dimensions $i \times n$, which contains the coefficients. These coefficients are indicative of the activity levels of each independent component within individual samples. Principal component analysis was performed on the covariance matrix between samples, after which the minimum number (representing i above) of top principal components that captured at least 85% of the total variance in the dataset was selected. The original mRNA expression level can be reconstructed by taking the inner product of the mixing matrix coefficients and the weights of the independent components for each gene. In ICA, an initial random weight vector with a variance of one must be chosen to achieve statistically independent components. Consequently, varying the initial random weight vectors can lead to different sets of independent components. To obtain a consensus set of independent components (referred to as TCs), 25 ICA runs were conducted, each with a unique random initialization weight vector.

Upon completion of all ICA runs, independent components with an absolute Pearson correlation coefficient greater than 0.9 were clustered, ensuring that the number of independent components in each cluster did not exceed the total number of ICA runs. TCs were derived by averaging the independent components within each cluster. Next, a credibility index for each TC was calculated by dividing the number of independent components in its cluster by the total number of ICA runs (25 in this case). TCs with a credibility index of 50% or higher were selected for constructing the TC matrix and the consensus mixing matrix.

By using bulk transcriptional profiles representing several healthy and disease conditions in the large intestine, we maximized our likelihood of identifying robust TCs that would capture the transcriptional patterns of as many distinct types of biological processes as possible that might be active in the large intestine in healthy tissue and cancerous tissue.

### Identifying the biological processes captured by the transcriptional components

We used multiple methods to identify the biological processes captured by the TCs (Fig. 1E). First, we conducted gene set enrichment analysis on all TCs using 13 gene set collections—including the Hallmark collection—sourced from the Molecular Signatures Database (MsigDB), version 7.1[20]. We included all gene sets comprising 10 – 500 genes after filtering out genes not present in the expression profiles of our integrated dataset. Enrichment for each gene set was evaluated using the two-sample Welch's t-test for unequal variance between the set of genes under investigation and the set of genes not under investigation. To compare gene sets of different sizes, we transformed Welch's t statistic into a Z-score. Second, we used the recently developed Transcriptional Adaptation to Copy Number Alterations (TACNA) profiling method[19]. This method served to identify TCs that capture the downstream effects of copy number alterations on gene expression levels. Third, we utilized the GenetICA-network (https://www.genetica-network.com)[21]. In brief, the GenetICA-network is an integrative method that predicts gene functions based on a guilt-by-association strategy utilizing more than 135,000 expression profiles. Using this method, we constructed co-functionality networks for the most important genes of each TC. Genes were deemed important in a TC if they had an absolute weight of 3 or greater within the TC. Next, the enrichment of predicted functionality was calculated for the resulting gene clusters. For continuous variables, we employed Spearman's rank correlation test. All statistical analyses were conducted using R version 3.6.2.

### Analysis of disease-free survival

To examine the association between the activity of various TCs and DFS, we used the subset of CRCs for which patient DFS data was available to perform univariate and multivariate Cox regression analyses (Fig. 1F). In the multivariate analysis, we used the following covariates: gender, microsatellite instability (MSI) status, v-Raf murine sarcoma viral oncogene homolog B (*BRAF*) status, Kirsten rat sarcoma virus (*KRAS*) status, primary tumor location, tumor stage, and treatment with adjuvant systemic therapy. The analysis was performed in a multivariate permutation framework with 10,000 permutations to control the false discovery rate at 5% with an 80% confidence level.

### Building the random survival forests

Separate RSFs were built for stage 2 and stage 3 colon cancer using TC activities as classifiers and DFS as the response variable (Fig. 1G). For each RSF, 1000 separate survival trees were built, each time selecting five randomly chosen TCs from the entire set of input TCs. The RSFs were built using a recursive process described in detail in Supplementary Materials.

For an RSF, all 1000 proximity matrices were summed element-wise to obtain a final proximity matrix. Next, hierarchical clustering using the Ward D2 method was performed on this final proximity matrix in order to identify patient subgroups of similarly classified patients based on the 1000 survival trees in the RSF. We then determined the maximum number of patient subgroups for which the log-rank test revealed a significant difference in the DFS curves.

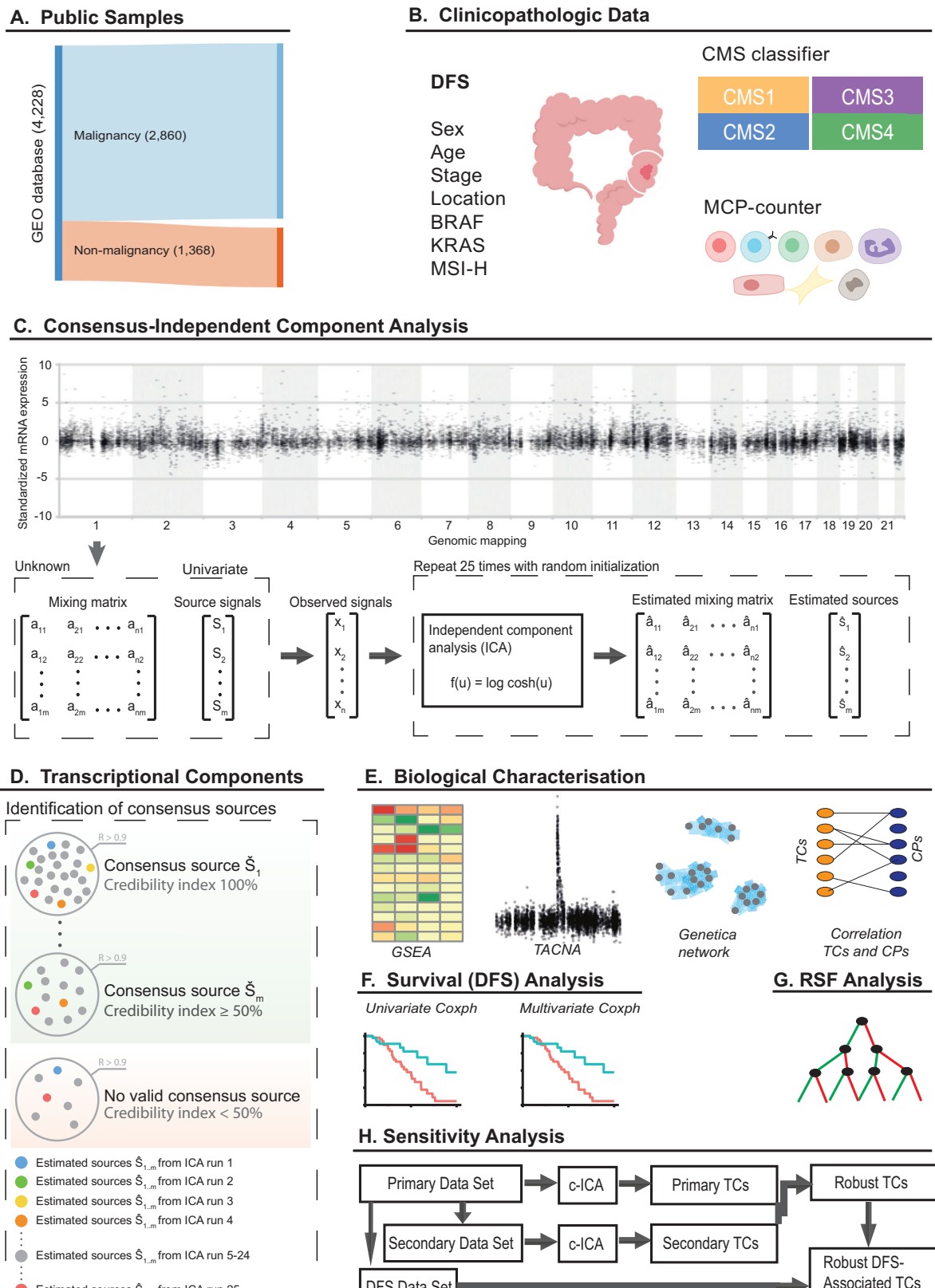

**A. Public Samples**

**B. Clinicopathologic Data**

**C. Consensus-Independent Component Analysis**

**D. Transcriptional Components**

**E. Biological Characterisation**

**F. Survival (DFS) Analysis**

**G. RSF Analysis**

**H. Sensitivity Analysis**

Biological interpretation of the RSFs was critical, as our goal was to determine how the combined activity of TCs that captured biological processes might lead to differences in DFS. Thus, for each RSF we calculated an "importance score" for each TC; this score reflects how often that TC is an important classifier in the 1000 survival trees in the forest.

**Spatial transcriptomics and single-cell transcriptome analysis**
To pinpoint the areas of significant TC activity in spatial transcriptomic profiles, we employed a permutation-based approach. We ran 5000 permutations for each TC-profile combination, yielding a p-value that indicates the extent to which the TC's activity in the corresponding profile differs

**Fig. 1 | Integrated approach to analyzing the transcriptional landscape of early colorectal cancer. A** An integrated data set of gene expression profiles was created from large intestine tissue samples was created. From the entire set, 2860 and 1368 samples were malignant and non-malignant tissue, respectively. **B** DFS, as well as other patient characteristics and clinicopathological information, were either collected or inferred. c-ICA was performed (**C**) in order to dissect the bulk transcriptomes into statistically independent TCs (**D**). **E** The various methods used to determine the biological processes captured by the TCs. **F** Idealized data showing how the putative association between TC activity and DFS was determined using univariate and multivariate Cox regression analyses. **G** RSF was performed in order to stratify patients with stage 2 and stage 3 colon cancer into distinct patient subgroups based on the patterns of TC activities. **H** A sensitivity analysis was performed in order to determine the robustness of the results. BRAF v-Raf murine sarcoma viral oncogene homolog B, c-ICA consensus independent component analysis, Coxph cox proportional regression, can copy number alteration, CMS consensus molecular subtype, CPs clinicopathological parameters, CRC colorectal cancer, DFS disease-free survival, GEO gene expression omnibus, GSEA gene set enrichment analysis, KRAS Kirsten rat sarcoma virus; 3, microenvironment cell population, MSI microsatellite instability, MSigDB molecular signatures database Hallmark gene set collection, RSF random survival forest, TACNA transcriptional adaptation to copy number alterations, TCs transcriptional components.

from what would be expected by chance (the null distribution). We then transformed these p-values into logarithmic values and represented them using a heatmap. This heatmap allowed us to visually explore the regions of the stained sample image where TC activity was significantly different from the expected random distribution. We used 3 publicly available spatial transcriptomic profiles, available at the 10x website (https://www.10xgenomics.com), in the spatial gene expression dataset.

Single-cell data in h5ad format were obtained from the Gut Cell Atlas (https://www.gutcellatlas.org/#datasets), encompassing a comprehensive single-cell RNA-seq dataset of 428,000 intestinal cells from fetal, pediatric, and adult donors across up to 11 distinct intestinal regions. Our analysis focused specifically on the mesenchyme lineage. We downloaded the normalized data and associated metadata for this lineage. We then implemented a sub-sampling strategy, randomly selecting 10% of cells from the mesenchyme lineage dataset, resulting in an analysis cohort of approximately 16,000 cells. For the projection analysis, we employed 3000 permutations with the Johnson transformation. To facilitate the visualization and interpretation of the corrected mixing matrix weights, box and whisker plots were created for each cell type within every TC panel.

### Sensitivity analysis
A sensitivity analysis was performed to determine the robustness of our results (Fig. 1H). In order to do so, we created a secondary dataset by excluding all samples from the primary dataset with available DFS data. The samples from the primary dataset that had annotated DFS were then added to a separate dataset, which we refer to as the DFS dataset.

To assess the robustness of the TCs, c-ICA was repeated on the secondary data set. Therefore, we defined TCs obtained from the primary data set as "primary TCs" and we defined TCs obtained from the secondary data set as "secondary TCs". To determine robustness, pairwise Pearson correlation coefficients were calculated between the gene weights of the primary and secondary TCs. Pairs of primary and secondary TCs with an absolute Pearson correlation coefficient >0.5 were considered robust and were therefore defined as robust TCs.

Next, we assessed the robustness of the associations between TC activities and DFS. First, we performed a cross-data set projection to determine the activity of the robust secondary TCs in the DFS data set. There was no overlap between the samples used to obtain the robust secondary TCs and the samples in the DFS data set; therefore, the DFS data set can be considered an independent data set in this analysis. Next, univariate and multivariate Cox regression analyses were performed on the DFS data set as described above in order to determine the associations between the activity of robust secondary TCs and DFS. TCs associated with DFS in the primary and independent DFS data sets were defined as "robust, DFS-associated TCs".

Finally, we assessed the robustness of the RSFs by performing the RSF building process twice, first using the DFS-associated primary TCs as input, and second using the robust DFS-associated secondary TCs as input. Robustness was then assessed by calculating the Pearson correlation coefficient between the resulting final proximity matrices.

### Reporting summary
Further information on research design is available in the Nature Portfolio Reporting Summary linked to this article.

## Results
### Identification of 191 robust, independent transcriptional components using c-ICA
After identification, pre-processing, removal of duplicates, and quality control based on samples obtained in 58 studies, a total of 4228 unique samples remained, including 2860 early CRC samples and 1368 non-malignant samples (Fig. 1A). We also removed all samples and studies with annotated DFS to create a secondary data set comprised of 1995 early CRC samples and all 1368 non-malignant samples for a sensitivity analysis (Supplementary Data 1).

c-ICA was subsequently used to dissect the bulk transcriptional profiles into statistically independent TCs (Fig. 1C and 1D), as described in the Methods and the Supplementary Methods. In short, c-ICA is a computational method to separate gene expression profiles into additive consensus transcriptional patterns (TCs) so that each TC is statistically as independent from the other TCs as possible. In every TC, each gene has a weight that describes how strongly and in which direction its expression level is influenced by a latent transcriptional regulatory factor. In the primary data set, we identified 220 independent primary TCs, of which 191 were robust, with a median absolute correlation coefficient of 0.92 (0.70-0.97; Supplementary Table 2) between pairs of primary and secondary TCs. The composition of these primary TCs and the gene set enrichment analysis (GSEA) results are publicly available and browsable by gene and gene set at our online portal available at http://transcriptional-landscape-colon.opendatainscience.net.

These primary TCs were found to be enriched for at least one gene set as defined in the 13 collections used. For example, 28% (61/220) of the primary TCs were enriched for at least one gene set from the Hallmark collection. The median top enrichment score (i.e., z-transformed p-value) for the Hallmark gene set collection for all 61 primary TCs was 7.24 (range: 4.07–28.06, IQR: 5.45–12.44). In addition, the transcriptional effects of copy number alterations (CNAs) were detected in 93 (42%) of the 220 primary TCs.

### Disease-free survival of patients with early-stage colorectal cancer was associated with the activity of 43 robust transcriptional components
Data regarding DFS were retrieved for 806 patients with colon cancer and 30 patients with rectal cancer. Patient characteristics and clinicopathological information are summarized in Table 1. We identified 53 DFS-associated primary TCs, of which 43 were robust. Figure 1E summarizes the methods used to obtain the biological characterization of the TCs. Figure 2 provides an overview of the biological processes enriched in the DFS-associated primary TCs. The TCs are ordered based on their association with DFS, represented as -log10(p-value). This figure indicates robust TCs with solid circles, while non-robust TCs are indicated with open circles. The left-hand side of Fig. 2 displays the results of GSEA focusing on Hallmark gene sets. What stands out is that many of the TCs most strongly associated with DFS were related to epithelial-mesenchymal transition (EMT), in particular, the following nine TCs demonstrated a strong association: TC208, TC117, TC193, TC202, TC116, TC153, TC58, TC136 and TC77 The chromosomal location of given CNAs, for which the TCs capture downstream effects

## Table 1 | Patient characteristics

| | DFS samples (N = 836 patients) | % |
|---|---|---|
| **Age (years)** | | |
| Median (range) | 69 (22–97) | NA |
| Unknown (N = 20) | NA | NA |
| **Sex** | | |
| Male | 440 | 53 |
| Female | 396 | 47 |
| **Stage** | | |
| <2 | 78 | 9 |
| 2 | 463 | 55 |
| 3 | 295 | 35 |
| **Location** | | |
| Proximal | 300 | 36 |
| Distal | 429 (30 rectal) | 51 |
| Unknown | 107 | 13 |
| **Adjuvant/neoadjuvant chemotherapy** | | |
| Yes | 209 (5 CRT) | 25 |
| No | 393 | 47 |
| Unknown | 234 | 28 |
| **Microsatellite status** | | |
| MSI-H | 73 | 9 |
| MSI-L or MSS | 356 | 43 |
| Unknown | 407 | 49 |
| **BRAF^v600E status** | | |
| Mutated | 37 | 4 |
| Wildtype | 360 | 43 |
| Unknown | 439 | 53 |
| **KRAS c12/13 status** | | |
| Mutated | 161 | 19 |
| Wildtype | 265 | 32 |
| Unknown | 410 | 49 |
| **CMS classification** | | |
| CMS1 | 98 | 12 |
| CMS2 | 136 | 16 |
| CMS3 | 110 | 13 |
| CMS4 | 249 | 30 |
| Unknown | 243 | 29 |

*BRAF* v-Raf murine sarcoma viral oncogene homolog B, *c*, codon, *CMS* consensus molecular subtype, *CRT* chemoradiotherapy, *DFS* disease-free survival, *KRAS* Kirsten rat sarcoma virus, *MSI-H* microsatellite instability high, *MSI-L* microsatellite instability low,; *MSS* microsatellite stable, *NA* not applicable.

on gene expression levels, is also presented in the middle. Transcriptional effects of gene CNAs were identified in 13 of these 43 robust TCs, four of which were specifically linked to chromosome 17q. Among these four TCs, TC148, TC149, and TC189 ranked fifth, seventh, and eighth in terms of their significance in association with DFS. On the right-hand side of Fig. 2, two heatmaps detail the direction and association between these DFS-associated TCs and various clinicopathological parameters. Notably, higher activity in many of the strongest DFS-associated TCs correlated with increased inferred abundance of specific cell types in the tumor microenvironment, particularly fibroblasts, endothelial cells, neutrophils, myeloid dendritic cells, monocytic lineage cells, and dendritic cells.

### Random survival forest analyses reveal patient subgroups with distinct patterns of biological process activities

Next, to identify potential patterns of TC activities that could be used to stratify patients with stage 2 and stage 3 colon cancer into distinct patient subgroups, we performed random survival forest (RSF) analyses (Fig. 1G). For these analyses only patients with colon cancer were included. RSF is a specialized statistical approach tailored for survival analysis. It is specifically advantageous for handling censored survival data while also offering predictive capabilities for time-to-event outcomes. The reasons for opting for RSF over alternative survival analysis methods lie in its inherent ability to manage complex datasets characterized by non-proportional hazards and nonlinear predictor-survival outcome relationships. We found that TC21 and TC117 had the highest importance scores (with scores of 0.99 and 1, respectively) for patients with stage 2 colon cancer ($n = 446$ patients) (Supplementary Table 2). The importance score reflects how often a TC is an important classifier in the 1000 survival trees in the forest. Higher TC21 activity was associated with the downregulation of genes enriched for the Hallmark gene set MYC targets, while higher TC117 activity was associated with the downregulation of genes enriched for the EMT and extracellular matrix (ECM) organization, and for the upregulation of genes enriched for KRAS signaling. Higher activity of TC21 was also associated with a higher inferred abundance of CD8 T cells. The final proximity matrix was robust (Pearson's $r = 0.9$).

RSF analysis of patients with stage 3 colon cancer ($n = 290$ patients) revealed that TC208 and TC77 had two of the highest importance scores, with scores of 1 and 0.95, respectively (Supplementary Table 2). Higher TC208 activity was associated with the upregulation of genes enriched for EMT, ECM, early and late estrogen response, KRAS signaling, and transforming growth factor β (TGF-β) signaling, while higher TC77 activity was associated with the upregulation of genes enriched for KRAS signaling, TGF-β signaling, the p53 pathway, apoptosis, and hypoxia and TNFα signaling via NF-κB and the inflammatory response. In addition, higher activities of both TC208 and TC77 were associated with a higher inferred abundance of several cell types in the tumor microenvironment. The final proximity matrix was robust (Pearson's $r = 0.68$).

For stage 2 and stage 3 colon cancer, a log-rank test revealed significant differences in the time course of DFS for up to three patient subgroups (Supplementary Figs. 1 and 2). However, utilizing patterns of TC activity, we stratified patients with stage 2 and stage 3 colon cancer into various biologically distinct subgroups. Although the sample size in this analysis was insufficient to reveal statistically significant differences in DFS among these subgroups, their unique patterns of biological processes may still hold clinical relevance. For instance, Fig. 3 illustrates the DFS curves for 10 patient subgroups identified by clustering the final proximity matrix specifically for patients with stage 2 colon cancer. These curves highlight the subgroups with the best and worst DFS outcomes. The corresponding clustering results can be found in Supplementary Fig. 3. The disparity in DFS between the best and worst survival subgroups can be largely attributed to variations in the activity levels of TC202, TC38, TC149, TC55, and TC147. Specifically, TC202 is characterized by a transcriptional pattern enriched for coagulation, apical junction, and EMT processes, and its activity is correlated with a higher inferred abundance of monocytic lineage, neutrophils, endothelial cells, and fibroblasts. TC38 captured the effects on gene expression levels due to CNAs at 17q and TC149 and TC55 captured the gene expression level effects of CNAs at 5q. Lastly, TC147 is linked with the upregulation of genes that are MYC targets and the downregulation of genes enriched for mitotic spindle and TGF-β signaling pathways.

### EMT associated DFS-associated TCs

In total nine DFS-associated TCs were identified that are associated with EMT (Fig. 2). These TCs were among the strongest associated with DFS and had high importance scores in the random survival forest analyses. Four of these nine TCs — TC202, TC116, TC153, and TC58 — had higher activity in samples classified as CMS4, as can be seen in the same figure. The aggregated activity profiles of these TCs across various samples are provided in

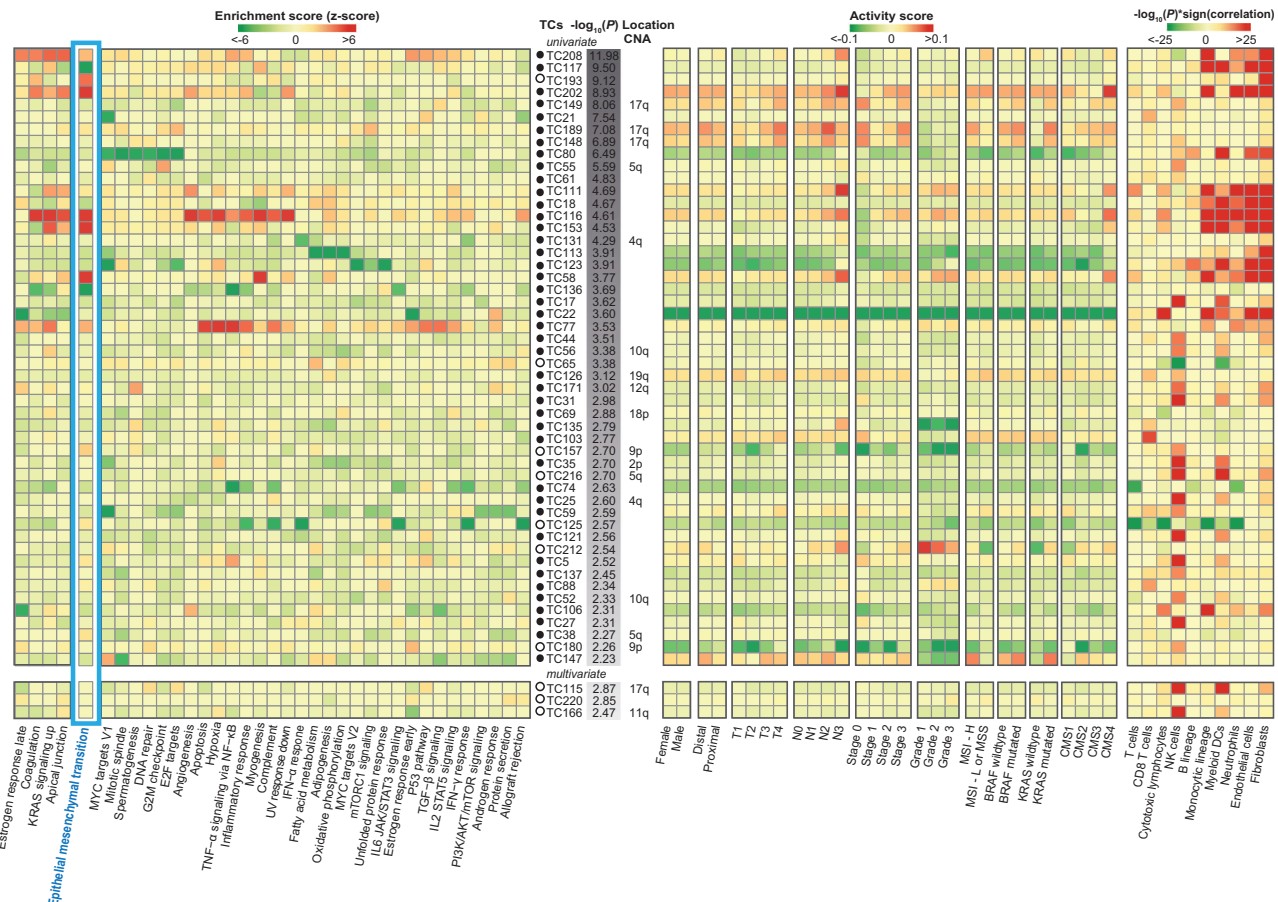

**Fig. 2 | Summary of the biological processes captured by transcriptional components associated with disease-free survival.** The 53 DFS-associated TCs based on the univariate or multivariate survival analysis are shown. The 43 robust TCs are annotated with solid circles, and the non-robust TCs are indicated with open circles. The TCs are shown in order of their association with DFS. The GSEA results of the 36 enriched Hallmark gene sets are shown on the left. Gene sets were included if the enrichment for at least one TC passed the Bonferroni threshold for multiple testing corrections. Gene sets were clustered using Spearman correlation and Ward D2. To facilitate the interpretation, heatmap colors are based on transformed $z$-scores and were truncated at an absolute value of 6. Red indicates enrichment of a biological process in the top-ranked genes in the TC (i.e., genes that are upregulated with a higher activity of the TC, and green indicates enrichment of a biological process in the bottom-ranked genes in the TC (i.e., genes that are downregulated with a higher activity of the TC). Thus, red and green indicate enriched biological processes comprising genes for which higher and lower expression, respectively, is associated with worse DFS. The chromosomal location of a given copy number alteration for which the TC captures the downstream effects on gene expression levels is shown in the middle. The activity scores per TC refer to the mean mixing matrix weights per

subgroup. The two heatmaps at the right show the association between TCs and CPs (CPs were included if the correlation for at least one TC passed the Bonferroni threshold for multiple testing correction). AKT Ak strain transforming, BRAF v-Raf murine sarcoma viral oncogene homolog B, CD8 cluster of differentiation, CNA,-copy number alteration, CMS consensus molecular subtypes, CP clinicopathological parameter, DFS disease-free survival, DNA deoxyribonucleic acid, E2F E2 factor, G2M gap2 mitosis, GSEA gene set enrichment analysis, IFN-α interferon-alfa, IFN-γ interferon-gamma, IL2 interleukin 2, IL6 interleukin 6, JAK Janus kinase, KRAS Kirsten rat sarcoma virus, MCP microenvironment cell population, MSI micro-satellite instability, MSI-H microsatellite instability high, MSI-L microsatellite instability low, MSS microsatellite stable, MtORC 1 mammalian target of rapamycin complex 1, MYC myelocytomatosis, NF-κB nuclear factor kappa B, NK natural killer, p53 protein 53, PI3K phosphatidylinositol 3-kinase, STAT3 signal transducer and activator of transcription 3, STAT5 signal transducer and activator of transcription 5, TCs transcriptional components, TGF-β transforming growth factor beta, TNF-α tumor necrosis factor alpha, TNM tumor node metastasis, UV ultraviolet.

Supplementary Fig. 4. This heatmap prominently showcases the heterogeneity in the activity of DFS-associated TCs related to EMT within the context of CRC samples, thereby highlighting the presence of distinct EMT-related biological processes operative in different instances of colorectal cancer.

Additionally, we analyzed spatial transcriptomic profiles from three colorectal cancer cases to scrutinize regions with significant activity of the EMT-related DFS-associated TCs. This analysis is visualized in Fig. 4 and Supplementary Fig. 5. Our data indicate a spatially variable distribution of these nine TCs' activities across different regions within each colorectal cancer tumor, particularly in the stromal compartments. This nuanced distribution underscores the inherent transcriptional heterogeneity of DFS-associated EMT-related processes, within individual tumor samples and across different tumor specimens.

To gain insight into the role of specific mesenchymal cells on the biology captured by the EMT-related, DFS-associated TCs, we utilized single-cell transcriptome data obtained from the Gut Cell Atlas (https://www.gutcellatlas.org/#datasets). This dataset is a comprehensive collection of single-cell RNA-seq profiles of 428,000 intestinal cells from fetal, pediatric, and adult donors, across up to 11 distinct intestinal regions. Supplementary Fig. 6 features box and whisker plots that illustrate the activity of the EMT-related DFS-associated TCs in various mesenchymal cell types. These plots provide valuable insights into the specific mesenchymal cell populations that might influence the biological processes associated with DFS. For example, mesothelium cells show pronounced higher activity of TC208, while pericytes show higher activity of TC117. Additionally, our analysis showed higher activity of TCs 202 and 136 in the stromal 4 cell type — a

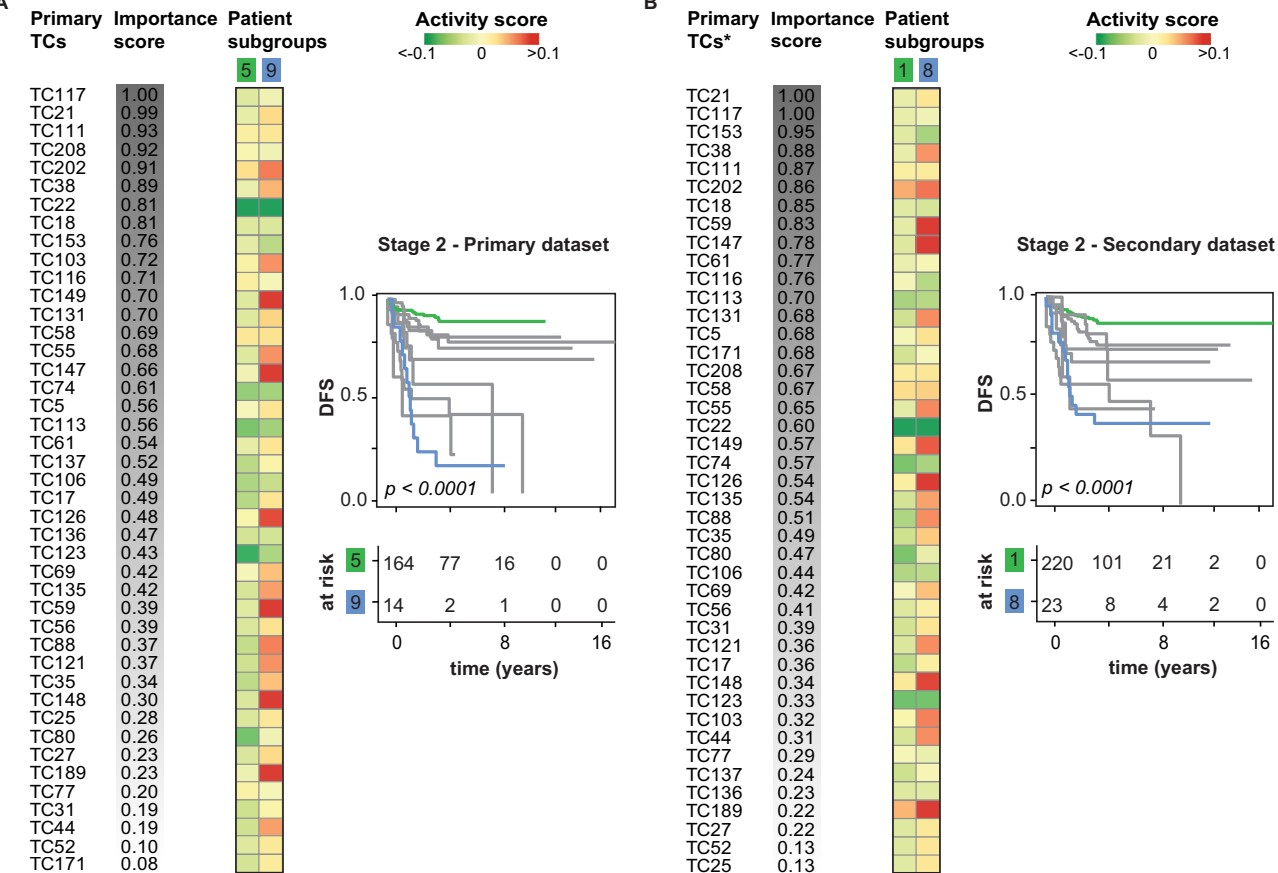

**Fig. 3 | Random survival forest analysis of stage 2 early colon cancer patients.** Shown are RSF analyses of 446 patients with stage 2 colon cancer based on the activity of 43 DFS-associated TCs as classifiers. RSF was performed as described in the Methods and Supplementary Materials. The RSF building process was performed twice, first with the DFS-associated primary TCs as the input (**A**), and then with the robust DFS-associated secondary TCs as the input (**B**). To simplify interpretation, the paired primary TCs are shown in **B** (indicated with an asterisk), rather than the secondary TCs; for information regarding the pairing of TCs, see Supplementary Table 2. For each RSF, an importance score was calculated for each TC, reflecting how often that TC is an important classifier in the survival trees in the forest. DFS disease-free survival, RFS random survival forest, TCs transcriptional components.

newly identified stromal population enriched for pro-inflammatory and fibroblastic reticular cell genes[22].

## Discussion

In this study, we identified a total of 191 robust transcriptional components that capture the transcriptional patterns of specific biological processes in early colorectal cancer. The activities of 43 of these robust TCs were associated with disease-free survival, enabling us to define patient subgroups based on distinct tumor-specific biological patterns.

Using our c-ICA–based approach of analyzing bulk transcriptomes, we identified extensive underlying transcriptional diversity among previously identified DFS-associated biological processes. A notable example is the epithelial-mesenchymal transition, for which we identified an association with nine independent, DFS-associated TCs, all with the coordinated upregulation and/or downregulation of various sets of genes and with heterogeneous activity within individual tumor samples and across different tumor specimens. Interestingly, EMT was previously identified as a key biological process associated with DFS in patients with early CMS4 CRC[11]. Strikingly, however, the activity of five of the nine EMT-related TCs was not associated with CMS4 CRC, indicating that EMT also plays a role in other tumor subtypes.

Previous studies have also attempted to refine the CMS classification in order to overcome the relative weakness of bulk analyzing expression profiles, which is strongly affected by the tumor stromal compartment and thus limits the detection of transcriptional patterns of more subtle biological processes[23,24]. These previous approaches focused on the tumor cells

themselves. In the first approach, CRC-intrinsic subtypes were identified by capturing colon cancer cell-specific gene expression patterns, although their prognostic value with respect to DFS was higher when they incorporated stromal compartment expression patterns in their analysis[23]. In the second approach, single-cell transcriptome analysis of colon cancer cells in 63 patients revealed the underlying tumor cell diversity and its association with DFS and identified a subset in which genes involved in EMT were upregulated[24]. Despite the value of these previous studies, we consider our c-ICA-based approach to bulk transcriptomes to be more informative, as it incorporates the transcriptomes of non-tumor cells but can still identify new, subtle DFS-associated transcriptional patterns.

In each TC, each gene has a "weight" that reflects how strongly and in which direction its expression level is affected by an underlying latent biological process. This enabled us to prioritize which genes may play key roles in the DFS-associated biological process that we identified. This is illustrated by TC208 and TC117. TC208 activity had both the strongest association with DFS and the highest importance score in the random survival forest analysis for stage 3 colon cancer. Higher activity was associated with the upregulation of genes involved in a transcriptional network activated by the *CDH1* gene, which encodes E-cadherin[25]. Upregulation of N-cadherin followed by downregulation of E-cadherin is a hallmark of EMT[26]. Higher TC208 activity was also strongly associated with increased expression of *KLK6* and *KLK10*, which encode kallikrein-related peptidase 6 and 10, respectively. Kallikreins can cleave E-cadherin proteins. Moreover, knocking down KLK10 in CRC cells has been shown to inhibit cell proliferation and induce apoptosis[27]. Kallikreins are potentially druggable[28–30].

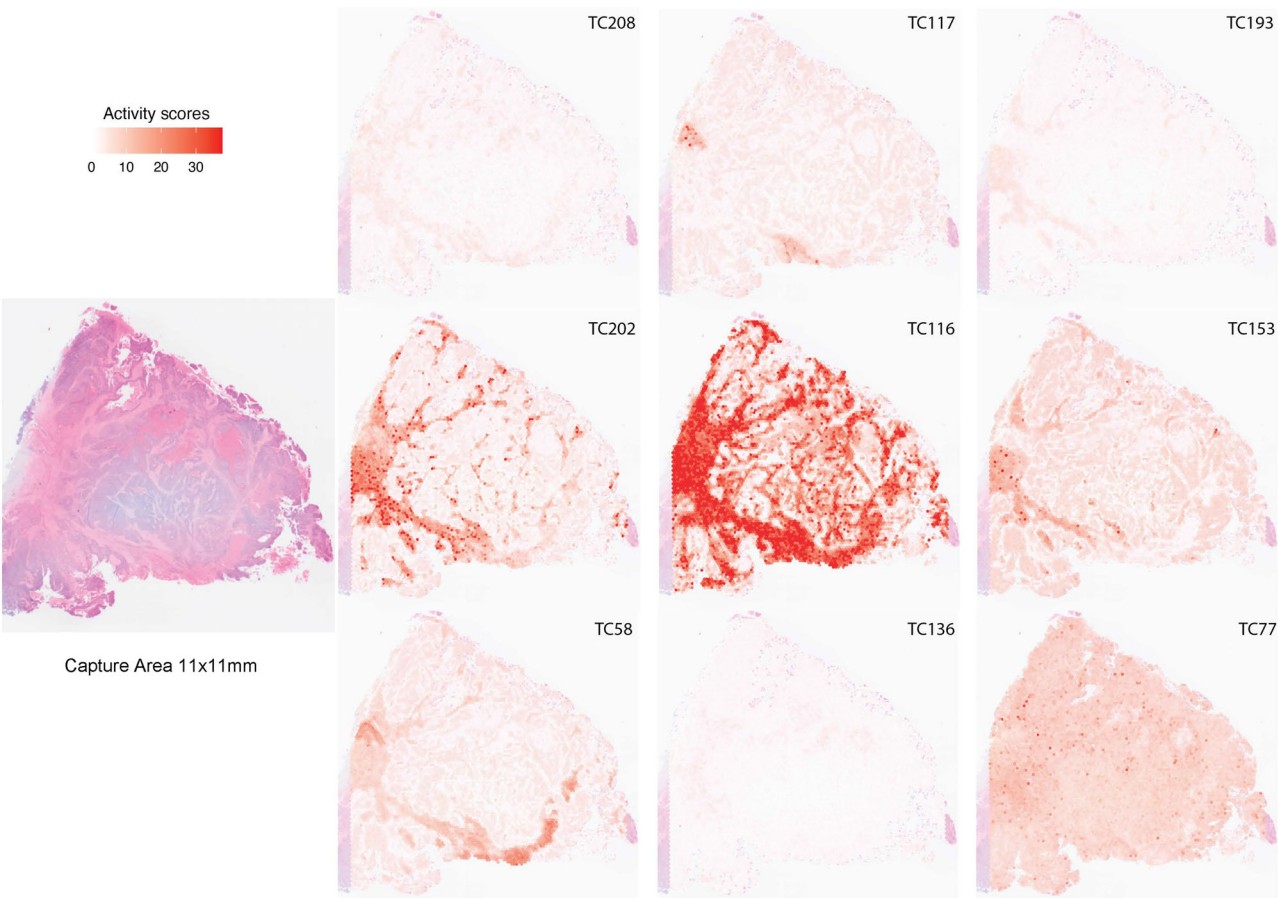

**Fig. 4 | Spatial transcriptomic profiles in colorectal cancer samples.** To pinpoint the areas of significant TC activity in spatial transcriptomic profiles, we employed a permutation-based approach. We ran 5000 permutations for each TC-profile combination, yielding a *p*-value that indicates the extent to which the TC's activity in the corresponding profile differs from what would be expected by chance (the null distribution). We then transformed these *p*-values into logarithmic values and represented them using a heatmap. In Fig. 4 the heatmap of one CRC sample is shown, in Supplementary Fig. 5, two more heatmap of CRC samples are shown. CRC colorectal cancer; TCs transcriptional components.

TC117 activity had the second-highest association with DFS in patients with CRC and the highest importance score for stage 2 colon cancer. High TC117 activity was associated with the downregulation of genes with strong enrichment for the *SNAI2* gene-based EMT signature[31]. Interestingly, the RNA gene *LINC00924* (Long Intergenic Non-Protein Coding RNA 924) was the most strongly upregulated gene, and higher *LINC00924* expression was associated with poor DFS in early CRC. Although little is known about its function, *LINC00924* expression has been previously shown to be associated with cell migration, cellular component movement, cell locomotion, and cell adhesion[32]. The activity of many DFS-associated robust TCs was, aside from EMT, also associated with ECM remodeling. For example, the high activity of TC202 and TC116 was associated with the upregulation of several matrix metalloproteinases and integrins, respectively. Numerous therapeutic agents, encompassing antibodies and small molecules targeting ECM components, are presently under development. Notably, approximately 90 integrin-based therapeutic drugs or imaging agents are in clinical trials, along with ongoing trials assessing matrix metalloproteinase inhibitors[33,34]. However, it is worth noting that conclusive evidence of therapeutic efficacy remains elusive. For instance, Andecaliximab, an inhibitor of matrix metalloproteinase 9, exhibited safety and demonstrated antitumor activity in a phase 1 trial. Nevertheless, its inclusion with FOL-FOX in a phase 3 trial for patients with advanced gastric or gastroesophageal junction adenocarcinoma did not yield improved overall survival outcomes[35].

In addition, 13 DFS-associated robust TCs captured the effect of CNAs on their respective mRNA levels. Aneuploidy and loss of heterozygosity are ubiquitous in microsatellite stable (MSS) CRCs, and CRCs become invasive and metastatic only when driver mutations co-occur with chromosomal instability[36,37]. Moreover, karyotyping to identify CNAs for risk stratification is routinely used in hematological malignancies[38]. Thus, our findings suggest that karyotyping may also help predict DFS more effectively in early CRC.

Our analysis is primarily hypotheses-generating. Moreover, due to the retrospective nature of our study, we were limited with respect to the availability of clinicopathological data. Thus, we could only robustly assess the prognostic value of TC activity, but could not assess its predictive value. The next logical step toward determining the clinical applicability of our findings would therefore be to perform the same analysis on prospectively collected data obtained during a randomized controlled clinical trial in an adjuvant therapy setting. Such a study could therefore address a research question based on one of the hypotheses generated here, such as whether TC activity can predict which patients are most likely to benefit from adjuvant therapy.

The composition of all primary TCs and our gene set enrichment analysis results are publicly available and browsable by gene and/or gene set at our online portal (http://transcriptional-landscape-colon. opendatainscience.net). Although we highlighted several biological processes and individual genes of interest here, more are present in our data. Indeed, we identified many biologically distinct patient subgroups in stage 2 and stage 3 colon cancer, suggesting that the current CMS classification of CRC could be adapted to be less restrictive by including the underlying transcriptional diversity of both pronounced and subtle biological processes. Although we did not find significant differences in DFS for all patient

subgroups, their diverse underlying biology suggests that each patient subgroup may need a specific systemic treatment strategy. We therefore invite researchers and clinicians to explore our data, thereby helping guide innovations in systemic treatment strategies.

## Data availability

Microarray expression data was collected from the public data repository: Gene Expression Omnibus with accession number GPL570 (generated with Affymetrix HG-U133 Plus 2.0). Single-cell data in h5ad format were obtained from the Gut Cell Atlas (https://www.gutcellatlas.org/#datasets). Three publicly available colorectal cancer spatial transcriptomic profiles were used obtained from the 10x website (https://www.10xgenomics.com). Specifically, https://www.10xgenomics.com/datasets/human-colorectal-cancer-11-mm-capture-area-ffpe-2-standard, https://www.10xgenomics.com/datasets/human-intestine-cancer-1-standard, and https://www.10xgenomics.com/datasets/human-colorectal-cancer-whole-transcriptome-analysis-1-standard-1-2-0. The composition of the primary transcriptional components and the gene set enrichment analysis results are publicly available and browsable by gene and gene set at our online portal available at http://transcriptional-landscape-colon.opendatainscience.net. Source data for the figures are available as Supplementary Data 2. For any other data inquiries, please contact the corresponding author.

## Code availability

The complete set of codes utilized in this study is available at a github repository[39].

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

## Acknowledgements
We thank the Center for Information Technology of the University of Groningen for their support and for providing access to the Peregrine high-performance computing cluster.

## Author contributions
D.G. Knapen: Conceptualization, resources, data curation, formal analysis, validation, investigation, visualization, methodology, writing—original draft, project administration and writing—review, and editing. S. Hone Lopez: Conceptualization, resources, data curation, formal analysis, investigation, visualization, methodology, project administration, and writing—review and editing. D.J.A. de Groot: Conceptualization, supervision, investigation, methodology, writing—review and editing. J.J. de Haan: Conceptualization, supervision, investigation, methodology, writing—review and editing. E.G.E. de Vries: Conceptualization, supervision, investigation, visualization, methodology, writing—review and editing. R. Dienstmann: Conceptualization, resources, investigation, methodology, project administration, writing—review and editing. S. de Jong: Conceptualization, supervision, investigation, visualization, methodology, writing—review and editing. A. Bhattacharya: Conceptualization, resources, data curation, software, formal analysis, supervision, validation, investigation, visualization, methodology, project administration, and writing—review and editing. R.S.N. Fehrmann: Conceptualization, resources, data curation, software, formal analysis, supervision, validation, investigation, visualization, methodology, project administration, and writing—review and editing.

## Competing interests
De Vries reports Institutional Financial Support for her advisory role from Crescendo Biologics, Daiichi Sankyo and NSABP, and Institutional Financial Support for clinical trials or contracted research from Amgen, AstraZeneca, Bayer, Crescendo Biologics, CytomX Therapeutics, G1 Therapeutics, Genentech, GE Healthcare, Regeneron, Roche and Servier, all outside the submitted work. De Haan reports Institutional Financial Support for clinical trials or contracted research from Astellas, Boehringer, Cogent, Incyte, Inhibrx, Zentalis, Zymeworks, all outside the submitted work. The other authors do not report any conflict of interest.
