## [Peer Review File · Communications Medicine]

Reviewers' comments:

Reviewer #1 (Remarks to the Author):

The manuscript by Knapen and colleagues “Independent transcriptional patterns reveal biological processes associated with disease-free survival in early colorectal cancer” aims to gain new insights in early CRC in order to develop more effective treatment strategies. To achieve that, the authors analysed 4,268 bulk-transcriptome data, downloaded from public database, and applied consensus independent component analysis (c-ICA), as describe in ref 35, and identified 191 robust transcriptional components (TCs). Out of this 191 TCs, 43 of them associate with disease-free survival (DFS). Furthermore, they also found 13 TCs related to copy number alterations (CNAs) at chromosome 17q. Lastly, they performed random survival forest (RSF) analysis and found that TC21 and TC117 have higher importance for patients with stage 2 colon cancer, while TC208 and TC77 have higher importance for patients with stage 3 colon cancer.

The manuscript is well written and the analysis fairly describe. However, the work is still highly descriptive and it is not immediately clear what novel biological insights are obtained nor new or specific treatment are offered. Either the clinical utility of their finding should be convincingly argued or novel insight in tumor biology should be provided. Hence, in my view the manuscript falls short in being a strong candidate for publication in this journal.

Please find some more detailed comment bellow:

1. In the abstract (L43-48), they mentioned that “early CRC may have eight different routes to achieve EMT, each requiring specific peri-operative treatment strategy”. However, the author did not discuss it further in the paper what kind of specific treatment can be given. They should either proposed concrete clinical strategy or novel insight in tumour biology about these eight routes should be provided.
2. The explanation in method section is not sufficient. For instance, it is unclear whether they used GSEApreranked in their analysis, if yes, how it was ranked, and what parameter was used. Similar problem can be found in other analysis such as TACNA, GenetICA-network, etc.
3. The figure legends could be improved. It is unclear what is the activity score in figure 2 mean and how it was calculated.
4. It is unclear to me which data supports their claim that TC148, TC149, and TC189 were associated most significantly with DFS among patients with early CRC. Is it because the activity score for stage 0 column is moderately high (red)? If yes, relatively similar level of activity can be found in stage 3 column. More explanations are needed.
5. It is also unclear how reliable these DFS associated TCs without further validations. The simplest validation one can do is to select top-n most important genes, based on weight, in each TCs and plot it as a heatmap in patient with good and/or bad DFS. Does it able to differentiate the good from bad DFS? Is there any similarity in expression level within group?
6. They also found certain TCs correlated with specific cell type. With the increasing number of single cell data publicly available, this finding should be easily validated. Do they really express in these specific cell type or is it an artifact of deconvolution process. This kind of validation is needed to

support other finding described in L120-122, L131-132, and L145-146.

7. For the 4 TCs-DFS that had transcriptional effects of CNAs at chromosome 17q, they can further validate this finding using the genotyping/WES/WGS data, which are publicly available. Is it true patients with high/low score TCs have 17q CNAs? If yes, more biological insight on why it may happen can also be discussed.

8. In the RFS analysis, it is not really clear to me what the importance score represent. Guessing from stage 2 results, it seems that higher activity of “important” TCs have favourable effect on DFS (downregulation of MYC, EMT, and ECM). However, on stage 3 results, higher activity of TC208 and TC77, which upregulate EMT, ECM, TGF β , P53, etc, are favourable for better DFS? More explanation or discussion are needed for results discussed in L114-133.

9. L134-140. The clustering results that yield 10 subgroups need to be shown, either as heatmap, TSNE, or UMAP plot, so reader can assess the goodness of those clusters. Furthermore, the biological importance of these 10 clusters also needs to be discussed.

10. L142-150. It is not clear why the author discussed the roles of TC202, TC38, TC149 and TC147 but ignoring TC103 and TC55, which has higher importance than TC147.

11. Out of these 4 TCs, TC202 has the highest importance score. However, this TCs is also related to CMS4 as well as an increase in the fibroblast abundance, which is already well known to have worst DFS (Guinney et al. 2015, Joanito et al. 2022). Hence, it is important to discuss further whether there is any additional new insight from their finding compared to previous studies.

12. Lastly, I think the author overclaimed their approach as written in L175-178. It is true that the gene weight can help identifying genes that are important in each TC, but how we can use this information in practice is not really straightforward. For example, for the top 50 genes, TC38 scores range from 5.14- 2.74, while TC202 scores range from 13.54-6.6. Hence, it is unclear, or at least not clearly demonstrated in the paper, why this method is better or more informative than conventional clustering followed by DEG analysis.

Reviewer #2 (Remarks to the Author):

In the manuscript by Knapen et al, entitled “Independent transcriptional patterns reveal biological processes associated with disease-free survival in early colorectal cancer” the authors outline the results obtained from a proposed new characterisation methodology that can identify subtle biological signalling from within bulk tumour transcriptomes. The authors details some of the findings, where they identify 191 transcriptional components (TCs) from 4000+ transcriptional profiles. Specifically, the authors indicate the presence of eight TCs within a previously homogeneous cascade characterised by EMT. This work offers a way to maximise the biological value of the large amount of existing bulk tumour data that exists, and as such adds a timely advancement to an important ongoing research area.

The experimental design appears sound however I feel that the way the paper is structured would benefit from a more detailed explanations and description of the processes underpinning the

methods prior to the results, and more detailed description of the Results individually as they are presented.

Overall, I fully appreciate that the authors have presented their method that they feel adds significant value to the field, and I also fully commend them on the fact they make their wide results publicly available/browsable, and their invitation to researchers/clinicians to explore it further.

The points raised below are primarily about trying to make the processes underpinning this method more intuitive and understandable for researchers in those fields. This should help it to appeal to the broader audience, therefore most of the points below are related to making the description/text within the Results section more reader-friendly.

Main essential points:

Point 1

Cohorts used: While there are certainly a lot of samples used here from across 58 studies, I believe that the inclusion of more details on the samples used at each point, through presentation of consort diagrams etc and collated tables, will help give the reader a bit more insight as to which samples were used for each of the analyses presented in the Results.

For example, line 99 indicates a set of 806 colon and 30 rectal.. were these part of the same cohorts described earlier, and if so how did they align to the other datasets? I think a consort diagram may make this a bit more easy to interpret.

Point 2

In the Introduction, lines 67-71, the authors state:

“consensus independent component analysis (c-ICA) can capture the transcriptional patterns of both robust and subtle biological processes by dissecting the tumor bulk transcriptomes into statistically independent components called transcriptional components (TCs). The activity of these TCs can then be determined in a bulk transcriptional profile.”

Which is the only real lead up to the results presented on line 84-85:

“In the primary data set, we identified 220 independent primary TCs, of which 191 were robust, with a median absolute correlation coefficient of 0.92 (0.70-0.97;”

As this paper is setting out this new method in this context, it would be appropriate to include far more detail on the steps involved here and the processes underlying the TC identification, what the meaning of “robust” is in this context and how it is calculated. As it stands, the sentence above tells the reader little about the method, how the results were arrived at, and what the results actually mean.

I appreciate that some of the Discussion section gives a lot of detail on this, but I think that these details needs to be fully outlined and explained at the start of the Results section.

I also appreciate that the authors include an extensive supplementary methods section, however I believe that these details need to be presented in the main figures in a comprehensive way. The authors state on line 232-233 that Supp Fig 1 contains a summary of the approach, however I don't see this in supp fig 1.

Point 3:

Results line 101 – similar to point above, hard to know how the 53 TCs were derived and how their robustness was measured.

Results line 102-103: “Transcriptional effects of gene CNAs were identified in 13 of these 43 robust TCs, of which four had transcriptional effects of CNAs at chromosome 17q.” Results line 105: “Fig. 2 summarizes these findings”

I’m unsure where and how Figure 2 demonstrates this; Can you more clearly articulate and signpost to where these results are in the Figures, by detailing each and every panel individually as the results are being described?

Point 4:

Results line 90-95: “These primary TCs were found to be enriched for at least one gene set as defined in the 13 collections used. For example, 28% (61/220) of the primary TCs were enriched for at least one gene set from the Hallmark collection. The median top enrichment score (i.e., z transformed p-value) for the Hallmark gene set collection for all 61 primary TCs was 7.24(range: 4.07–28.06, IQR: 5.45–12.44). In addition, the transcriptional effects of copy number alterations (CNAs) were detected in 93 (42%) of the 220 primary TCs.”

I was quite lost as to where each of these individual and specific points relate to in the figures provided, and quite how they were arrived at. I have no doubt that this method is drawing out new and important information, but given the amount of information presented in Figure 1 alone, there needs to be far more detailed alignment between the text and the figures. Currently the structure of the paper means that this section and its relationship with Figure 1 is hard to interpret.

Minor:

Line 111-112 – “we performed random survival forest (RSF) analyses” – what is RSF and why was it chosen over other more standard methods?

Line 114 – “We found that TC21 and TC117 had the highest importance scores” – what are the importance scores, how are they derived?

Overall, I really do value this story and hope that some additional detail to guide the reader through the process and results will help it to become a more utilised methods in future studies.

Letter addressing changes made in COMSMED-23-0299-T entitled "Independent transcriptional patterns reveal biological processes associated with disease-free survival in early colorectal cancer"

We would like to thank both reviewers for their insightful comments, which have helped us further improve our manuscript.

Responses to remarks from reviewers

Reviewer #1

The manuscript by Knapen and colleagues "Independent transcriptional patterns reveal biological processes associated with disease-free survival in early colorectal cancer" aims to gain new insights in early CRC in order to develop more effective treatment strategies. To achieve that, the authors analysed 4,268 bulk-transcriptome data, downloaded from public database, and applied consensus independent component analysis (c-ICA), as describe in ref 35, and identified 191 robust transcriptional components (TCs). Out of this 191 TCs, 43 of them associate with disease-free survival (DFS). Furthermore, they also found 13 TCs related to copy number alterations (CNAs) at chromosome 17q. Lastly, they performed random survival forest (RSF) analysis and found that TC21 and TC117 have higher importance for patients with stage 2 colon cancer, while TC208 and TC77 have higher importance for patients with stage 3 colon cancer.

The manuscript is well written and the analysis fairly describe. However, the work is still highly descriptive and it is not immediately clear what novel biological insights are obtained nor new or specific treatment are offered. Either the clinical utility of their finding should be convincingly argued or novel insight in tumor biology should be provided. Hence, in my view the manuscript falls short in being a strong candidate for publication in this journal.

Please find some more detailed comment bellow:

Remark 1.

In the abstract (L43-48), they mentioned that "early CRC may have eight different routes to achieve EMT, each requiring specific peri-operative treatment strategy". However, the author did not discuss it further in the paper what kind of specific treatment can be given. They should either proposed concrete clinical strategy or novel insight in tumour biology about these eight routes should be provided.

We concur with the reviewer's suggestion for a more comprehensive explanation. To address this, we have incorporated a heatmap that illustrates the activity scores of the Epithelial-Mesenchymal Transition (EMT)-related Transcriptional Components (TCs) within colorectal cancer (CRC) samples. This heatmap highlights the heterogeneity in TC activity across colorectal cancer samples, signifying that distinct biological processes related to EMT are at play in different CRC samples. The heatmap can be found on page 2 of the rebuttal letter, and in high resolution as Supplementary Fig. S4. Consequently, this underscores the notion that targeting EMT may require different treatment strategies for different patients. Of note, we ultimately identified nine DFS-associated TCs that were associated with EMT instead of eight.

In the text, this now reads: “In total nine DFS-associated TCs were identified that are associated with EMT (Fig. 2). These TCs were among the strongest associated with DFS and had high importance scores in the random survival forest analyses. Four of these nine TCs — TC202, TC116, TC153, and TC58 — had higher activity in samples classified as CMS4, as can be seen in the same figure. The aggregated activity profiles of these TCs across various samples are provided in Supplementary Fig. S4. This heatmap prominently showcases the heterogeneity in the activity of DFS-associated TCs related to EMT within the context of CRC samples, thereby highlighting the presence of distinct EMT-related biological processes operative in different instances of colorectal cancer.” Page 15, lines 6-14.

We have emphasized in the discussion that our approach is capable of gaining insight into the biology of these EMT-related TCs, which might be a step forward to novel treatment strategies. We have illustrated this with TC208 and TC117. This text reads:

“This is illustrated by TC208 and TC117. TC208 activity had both the strongest association with DFS and the highest importance score in the random survival forest analysis for stage 3 colon cancer. Higher activity was associated with the upregulation of genes involved in a transcriptional network activated by the CDH1 gene, which encodes E-cadherin.²⁴ Upregulation of N-cadherin followed by downregulation of E-cadherin is a hallmark of EMT.²⁵ Higher TC208 activity was also strongly associated with increased expression of KLK6 and KLK10, which encode kallikrein-related peptidase 6 and 10, respectively. Kallikreins can cleave E-cadherin proteins. Moreover, knocking down KLK10 in CRC cells has been shown to inhibit cell proliferation and induce apoptosis.²⁶ Kallikreins are potentially druggable.²⁷⁻²⁹ TC117 activity had the second-highest association with DFS in patients with CRC and the highest importance score for stage 2 colon cancer. High TC117 activity was associated with the downregulation of genes with strong enrichment for the SNAI2 gene based EMT signature.³⁰ Interestingly, the RNA gene LINC00924 (Long Intergenic Non-Protein Coding RNA 924) was the most strongly upregulated gene, and higher LINC00924 expression was associated with poor DFS in early CRC. Although little is known about its function, LINC00924 expression has been previously shown to be associated with cell migration, cellular component movement, cell locomotion, and cell adhesion.³¹ The activity of many DFS-associated robust TCs was, aside from EMT, also associated with ECM remodeling. For example, the high activity of TC202 and TC116 was associated with the upregulation of several matrix metalloproteinases and integrins, respectively. Numerous therapeutic agents, encompassing antibodies and small molecules targeting ECM components, are presently under development. Notably, approximately 90 integrin-based therapeutic drugs or imaging agents are in clinical trials, along with ongoing trials assessing matrix metalloproteinase inhibitors.^{32,33} However, it is worth noting that conclusive evidence of therapeutic efficacy remains elusive. For instance, Andecaliximab, an inhibitor of matrix metalloproteinase 9, exhibited safety and demonstrated antitumor activity in a phase 1 trial. Nevertheless, its inclusion with FOLFOX in a phase 3 trial for patients with advanced gastric or gastroesophageal junction adenocarcinoma did not yield improved overall survival outcomes.³⁴” Page 17, lines 9-24 and page 18, lines 1-14.

Remark 2.

The explanation in method section is not sufficient. For instance, it is unclear whether they used GSEA preranked in their analysis, if yes, how it was ranked, and what parameter was used. Similar problem can be found in other analysis such as TACNA, GenetICA-network, etc.

We agree with the reviewer and have now included additional information on the methods used. This now reads:

“We used multiple methods to identify the biological processes captured by the TCs. First, we conducted gene set enrichment analysis on all TCs using 13 gene set collections — including the Hallmark collection — sourced from the Molecular Signatures Database (MsigDB), version 7.1.³⁶ We included all gene sets comprising 10 – 500 genes after filtering out genes not present in the expression profiles of our integrated dataset. Enrichment for each gene set was evaluated using the two-sample Welch's t-test for unequal variance between the set of genes under investigation and the set of genes not under investigation. To compare gene sets of different sizes, we transformed Welch's t statistic into a Z-score. Second, we used the recently developed Transcriptional Adaptation to Copy Number Alterations (TACNA) profiling method.³⁵ This method served to identify TCs that capture the downstream effects of copy number alterations on gene expression levels. Third, we utilized the GenetICA-network (<https://www.genetica-network.com>).³⁷ In brief, the GenetICA-network is an integrative method that predicts gene functions based on a guilt-by-association strategy utilizing more than 135,000 expression profiles. Using this method, we constructed co-functionality networks for the most important genes of each TC. Genes were deemed important in a TC if they had an absolute weight of 3 or greater within the TC. Next, the enrichment of predicted functionality was calculated for the resulting gene clusters. For continuous variables, we employed Spearman's rank correlation

test. All statistical analyses were conducted using R version 3.6.2.” Page 6, lines 10-24, and page 7, lines 1-4.

Furthermore, we added the following: “The pre-processed and normalized expression profiles were used to infer the tumor microenvironment composition using the Microenvironment Cell Population (MCP) counter, version 1.1, available as an R package.³⁴ The MCP counter estimates the absolute abundance of eight immune and two stromal cell populations on bulk gene expression profiles. We utilized the random forest CMS classifier to assign CMS subtypes to the CRCs.¹¹” Page 5, lines 19-23.

Remark 3.

The figure legends could be improved. It is unclear what is the activity score in figure 2 mean and how it was calculated.

This has been added and reads: “The activity scores of each TC refer to the mean mixing matrix weights per subgroup.” Page 32, lines 15-16.

Remark 4.

It is unclear to me which data supports their claim that TC148, TC149, and TC189 were associated most significantly with DFS among patients with early CRC. Is it because the activity score for stage 0 column is moderately high (red)? If yes, relatively similar level of activity can be found in stage 3 column. More explanations are needed.

It is indeed true that our previous statement was unclear, and we have now made a correction. TC148, TC149, and TC189 are not the TCs most significantly associated with DFS; rather, they are the fifth, seventh, and eighth most significantly associated TCs with DFS, respectively. This ranking can be visually confirmed in Fig. 2, where the TCs are presented in order starting from the most to the least associated with DFS.

This revised text now reads: “Transcriptional effects of gene CNAs were identified in 13 of these 43 robust TCs, four of which were specifically linked to chromosome 17q. Among these four TCs, TC148, TC149, and TC189, ranked fifth, seventh, and eighth in terms of their significance in association with DFS.” Page 11, lines 17-20.

Remark 5.

It is also unclear how reliable these DFS associated TCs without further validations. The simplest validation one can do is to select top-n most important genes, based on weight, in each TCs and plot it as a heatmap in patient with good and/or bad DFS. Does it able to differentiate the good from bad DFS? Is there any similarity in expression level within group?

In Independent Component Analysis, it is not uncommon for specific genes to appear as important contributors in multiple independent components (i.e., transcriptional components; TCs). This occurs because each TC represents a distinct biological process or pattern, and individual genes can often be involved in multiple biological pathways simultaneously.

When analyzing bulk transcriptomes, it is essential to recognize that the measured expression level of a particular gene is often the cumulative result of multiple biological processes at work. Some of these processes may be subtle yet highly relevant for

DFS in CRC. The challenge in bulk transcriptome analysis is that these subtle signals can easily be masked or overshadowed by more pronounced biological processes that, while significant in terms of their impact on gene expression, may not necessarily be pertinent to DFS.

By relying solely on the expression level of a single gene in bulk transcriptomes, one risks missing out on these subtle signals that could be vital for understanding survival outcomes. In a bulk setting, the specific signal of interest could become indiscernible as it gets swamped by the noise from other, more dominant, but not necessarily DFS-relevant, biological processes.

Therefore, a comprehensive approach, like c-ICA, that considers the interplay of multiple genes within each independent component offers a more nuanced view. It allows for the detection of subtler yet profoundly important, biological processes directly bearing on DFS in colorectal cancer.

Furthermore, we performed an extensive sensitivity analysis to determine the robustness of our results, which is described in the manuscript. For clarity, we also provide the method described in the manuscript here.

“We created a secondary dataset by excluding all samples with available DFS data from the primary dataset. To assess the robustness of the TCs, c-ICA was repeated on the secondary data set. Therefore, we defined TCs obtained from the primary data set as ‘primary TCs’ and the TCs obtained from the secondary data set as ‘secondary TCs’. To evaluate robustness, pairwise Pearson correlation coefficients were calculated between the gene weights of the primary and secondary TCs. Pairs of primary and secondary TCs with an absolute Pearson correlation coefficient greater than 0.5 were considered robust and thus defined as robust TCs. Next, we assessed the robustness of the associations between TC activities and DFS. First, we performed a cross-data set projection to determine the activity of the robust secondary TCs in the DFS data set. There was no overlap between the samples used to obtain the robust secondary TCs and the samples in the DFS data set; therefore, the DFS data set can be considered an independent data set in this analysis. Next, univariate and multivariate Cox regression analyses were performed on the DFS data set as described above in order to determine the associations between the activity of robust secondary TCs and DFS. TCs associated with DFS in the primary and independent DFS data sets were defined as “robust, DFS-associated TCs”. Finally, we assessed the robustness of the RSFs by performing the RSF building process twice, first using the DFS-associated primary TCs as input, and second using the robust DFS-associated secondary TCs as input. Robustness was then assessed by calculating the Pearson correlation coefficient between the resulting final proximity matrices.”

Remark 6.

They also found certain TCs correlated with specific cell type. With the increasing number of single cell data publicly available, this finding should be easily validated. Do they really express in these specific cell type or is it an artifact of deconvolution process. This kind of validation is needed to support other finding described in L120-122, L131-132, and L145-146.

To dispel any concerns that the TCs might be artifacts or products of a deconvolution

process, we analyzed spatial transcriptomic profiles in three distinct cases of colorectal cancer, focusing specifically on the EMT-associated TCs. Through this analysis, we found that the activity of these TCs differs across various regions within colorectal cancer tumors and their corresponding stromal compartments. This observation underscores the presence of these biological processes in different cell types. In the manuscript, this is articulated as follows:

“Additionally, we analyzed spatial transcriptomic profiles from three colorectal cancer cases to scrutinize regions with significant activity of the EMT-related DFS-associated TCs. This analysis is visualized in Fig. 4 and Supplementary Fig. S5. Fig. 4 can be seen below. Our data indicate a spatially variable distribution of these nine TCs’ activities across different regions within each colorectal cancer tumor, particularly in the stromal compartments. This nuanced distribution underscores the inherent transcriptional heterogeneity of DFS-associated EMT-related processes, within individual tumor samples and across different tumor specimens.” Page 15, lines 15-22.

Figure 2

Remark 7.

For the 4 TCs-DFS that had transcriptional effects of CNAs at chromosome 17q, they can further validate this finding using the genotyping/WES/WGS data, which are publicly available. Is it true patients with high/low score TCs have 17q CNAs? If yes, more biological insight on why it may happen can also be discussed.

In a previous publication, we detailed a platform-independent approach known as transcriptional adaptation to CNA profiling (TACNA profiling). This technique isolates the transcriptional effects of CNA from gene bulk expression profiles, eliminating the need for paired CNA profiles. In that publication, we conducted a comprehensive

analysis; we obtained CNA profiles from SNP arrays for a subset of samples in the TCGA dataset (n = 10,620). TACNA profiles demonstrated clear alignment with their corresponding, independently generated CNA profiles (Bhattacharya, A. et al. Transcriptional effects of copy number alterations in a large set of human cancers. Nat Commun 2020; 11, 715).

We extracted TCGA samples that exhibited distinct chromosome 17 CNAs to further validate our current findings. Utilizing the four TCs that captured these chromosome 17 CNAs, we generated a TACNA profile for the TCGA samples harboring a chromosome 17 CNA. TACNA levels are indicated in black in the resulting visual representation, while the red dots represent the retrieved CNA profiles derived from SNP arrays. The observed patterns correspond with one another.

Remark 8.

In the RFS analysis, it is not really clear to me what the importance score represent. Guessing from stage 2 results, it seems that higher activity of “important” TCs have favourable effect on DFS (downregulation of MYC, EMT, and ECM). However, on

stage 3 results, higher activity of TC208 and TC77, which upregulate EMT, ECM, TGFB, P53, etc, are favourable for better DFS? More explanation or discussion are needed for results discussed in L114-133.

In the paper's methods section, we initially included the following explanation: "Thus, for each RSF, we calculated an "importance score" for each TC; this score reflects how often that TC is an important classifier in the 1000 survival trees in the forest." We have now included this clarification in the results section as well.

Remark 9.

L134-140. The clustering results that yield 10 subgroups need to be shown, either as heatmap, TSNE, or UMAP plot, so reader can assess the goodness of those clusters. Furthermore, the biological importance of these 10 clusters also needs to be discussed.

We have now incorporated a heatmap to display the clustering results that identified 10 distinct subgroups. The heatmap can be found on page 9 and in high resolution as Supplementary Fig. S3. The revised text, which also elaborates on the biological importance of these clusters, reads as follows: "However, utilizing patterns of TC activity, we stratified patients with stage 2 and stage 3 colon cancer into various biologically distinct subgroups. Although the sample size in this analysis was insufficient to reveal statistically significant differences in DFS among these subgroups, their unique patterns of biological processes may still hold clinical relevance. For instance, Fig. 3 illustrates the DFS curves for 10 patient subgroups identified by clustering the final proximity matrix specifically for patients with stage 2 colon cancer. These curves highlight the subgroups with the best and worst DFS outcomes. The corresponding clustering results can be found in Supplementary Fig S3. The disparity in DFS between the best and worst survival subgroups can be largely attributed to variations in the activity levels of TC202, TC38, TC149, TC55, and TC147. Specifically, TC202 is characterized by a transcriptional pattern enriched for coagulation, apical junction, and EMT processes, and its activity is correlated with a higher inferred abundance of fibroblasts. TC38 captured the effects on gene expression levels due to CNAs at 17q, and TC149 and TC55 captured the gene expression level effects of CNAs at 5q. Lastly, TC147 is linked with the upregulation of genes that are MYC targets and the downregulation of genes enriched for mitotic spindle and TGF- β signaling pathways. This TC is also associated with a higher inferred abundance of natural killer (NK) cells." Page 13, lines 20-24, and page 14, lines 1-18.

Remark 10.

L142-150. It is not clear why the author discussed the roles of TC202, TC38, TC149 and TC147 but ignoring TC103 and TC55, which has higher importance than TC147.

We acknowledge the oversight and appreciate the opportunity to clarify our selection criteria for discussing TCs. We initially chose to focus on TCs with the most notable differences in activity between the best and worst survival subgroups and that also had high importance scores in the random survival forest (RSF) analyses conducted using both the primary and secondary datasets. Although TC103 displayed a high importance score in the RSF of the primary dataset, it had a considerably lower score in the analysis of the secondary dataset, which is why it was not included. However, we have now incorporated a discussion of TC 55, which also showed high importance. The revised text is included in the answer to remark 9.

Remark 11.

Out of these 4 TCs, TC202 has the highest importance score. However, this TCs is also related to CMS4 as well as an increase in the fibroblast abundance, which is already well known to have worst DFS (Guinney et al. 2015, Joanito et al. 2022). Hence, it is important to discuss further whether there is any additional new insight from their finding compared to previous studies.

While TC 202 does indeed capture elements of EMT and is associated with CMS4, this alone does not provide new insight. However, we also identified three additional DFS-associated, EMT-related TCS – TC116, TC153, and TC58 — that exhibited higher activity in CMS4 samples. These four TCs revealed distinct patterns of coordinated gene upregulation and/or downregulation, showcasing heterogeneous activity within individual tumor samples and across various tumor specimens. This variability highlights the intricate transcriptional heterogeneity within the CMS4

subtype of colorectal tumors. Notably, five out of the nine DFS-associated, EMT-related TCs did not show higher activity in CMS4 samples, suggesting that EMT processes are not confined to the CMS4 subtype and may play a significant role in other tumor subtypes as well.

Remark 12.

Lastly, I think the author overclaimed their approach as written in L175-178. It is true that the gene weight can help identifying genes that are important in each TC, but how we can use this information in practice is not really straightforward. For example, for the top 50 genes, TC38 scores range from 5.14- 2.74, while TC202 scores range from 13.54-6.6. Hence, it is unclear, or at least not clearly demonstrated in the paper, why this method is better or more informative than conventional clustering followed by DEG analysis.

We appreciate the reviewer's concern regarding the practical application of our method as compared to conventional clustering followed by DEG analysis. However, it's crucial to recognize that DEG analysis often overlooks subtle but biologically important signals, and instead favor more pronounced gene expression changes. For a more detailed rationale on why we believe exploring the differential activity of Transcriptional Components (TCs) is crucial, rather than solely focusing on differential gene expression in bulk transcriptomes, we refer to our response to Remark 5 from Reviewer 1.

For practical interpretation of a TC showing differential activity associated with DFS, we recommend focusing on 'outlier genes', defined as those with absolute weight values above 3. The composition of these TCs, along with the gene set enrichment analysis (GSEA) results, are publicly accessible and searchable by gene and gene set at our online portal available at <http://transcriptional-landscape-colon.opendatainscience.net>.

Additionally, the directionality of weights within a TC offers valuable insights into the underlying biology. Specifically, genes with positive weights are inversely related to those with negative weights within the biological process captured by that TC. This directional information augments our comprehension of regulatory dynamics within TCs, offering a richer understanding of the complex transcriptional mechanisms involved in various biological processes.

Reviewer #2

In the manuscript by Knapen et al, entitled "Independent transcriptional patterns reveal biological processes associated with disease-free survival in early colorectal cancer" the authors outline the results obtained from a proposed new characterisation methodology that can identify subtle biological signalling from within bulk tumour transcriptomes. The authors details some of the findings, where they identify 191 transcriptional components (TCs) from 4000+ transcriptional profiles. Specifically, the authors indicate the presence of eight TCs within a previously homogeneous cascade characterised by EMT. This work offers a way to maximise the biological value of the large amount of existing bulk tumour data that exists, and as such adds a timely advancement to an important ongoing research area.

The experimental design appears sound however I feel that the way the paper is structured would benefit from a more detailed explanations and description of the processes underpinning the methods prior to the results, and more detailed description of the Results individually as they are presented.

Overall, I fully appreciate that the authors have presented their method that they feel adds significant value to the field, and I also fully commend them on the fact they make their wide results publicly available/browsable, and their invitation to researchers/clinicians to explore it further.

The points raised below are primarily about trying to make the processes underpinning this method more intuitive and understandable for researchers in those fields. This should help it to appeal to the broader audience, therefore most of the points below are related to making the description/text within the Results section more reader-friendly.

Main essential points:

Remark 1.

Cohorts used: While there are certainly a lot of samples used here from across 58 studies, I believe that the inclusion of more details on the samples used at each point, through presentation of consort diagrams etc and collated tables, will help give the reader a bit more insight as to which samples were used for each of the analyses presented in the Results. For example, line 99 indicates a set of 806 colon and 30 rectal. Were these part of the same cohorts described earlier, and if so how did they align to the other datasets? I think a consort diagram may make this a bit more easy to interpret.

We completely agree with the reviewer's suggestion for more clarity on the samples used in each analysis. To address this, we have added a supplementary table (supplementary table S1) that explicitly lists which samples were used in which analyses. This should provide a clearer overview and assist in interpreting our results.

Remark 2.

In the Introduction, lines 67-71, the authors state: "consensus independent component analysis (c-ICA) can capture the transcriptional patterns of both robust and subtle biological processes by dissecting the tumor bulk transcriptomes into statistically independent components called transcriptional components (TCs). The activity of these TCs can then be determined in a bulk transcriptional profile."

Which is the only real lead up to the results presented on line 84-85: "In the primary data set, we identified 220 independent primary TCs, of which 191 were robust, with a median absolute correlation coefficient of 0.92 (0.70-0.97;"

As this paper is setting out this new method in this context, it would be appropriate to include far more detail on the steps involved here and the processes underlying the TC identification, what the meaning of "robust" is in this context and how it is calculated. As it stands, the sentence above tells the reader little about the method, how the results were arrived at, and what the results actually mean.

I appreciate that some of the Discussion section gives a lot of detail on this, but I think that these details need to be fully outlined and explained at the start of the Results section.

I also appreciate that the authors include an extensive supplementary methods section, however I believe that these details need to be presented in the main figures in a comprehensive way.

We fully agree with the reviewer. We have moved the methods section to precede the results section to improve the manuscript's structure and clarity. Additionally, we have included a new paragraph in the results section to elucidate our use of c-ICA for extracting statistically independent TCs from the bulk transcriptional profiles. The added text reads:

“c-ICA was subsequently used to dissect the bulk transcriptional profiles into statistically independent TCs, as described in the Methods and the Supplementary Methods. In short, c-ICA is a computational method to separate gene expression profiles into additive consensus transcriptional patterns (TCs) so that each TC is statistically as independent from the other TCs as possible. In every TC, each gene has a weight that describes how strongly and in which direction its expression level is influenced by a latent transcriptional regulatory factor.” Page 10, lines 9-15.

We now have detailed information on how we identify biological processes captured by each TC. We refer to the response to Reviewer 1 Remark 2 for more information on this.

Remark 3.

The authors state on line 232-233 that Supp Fig 1 contains a summary of the approach, however I don't see this in supp fig 1.

It is correct that the overview of our data analysis approach is in Fig. 1, not in Supplementary Fig. 1, as previously indicated. To clarify this, we have revised the sentence, which now reads:

“Detailed information regarding the methods is provided in the Supplementary Materials. An overview of our data analysis approach is presented in Fig. 1.” Page 5, lines 5-6.

Remark 4.

Results line 101 – similar to point above, hard to know how the 53 TCs were derived and how their robustness was measured.

We think with the restructuring of the manuscript mentioned this is now clearer. Please see the response to Reviewer 2 Remark 2.

Remark 5.

Results line 102-103: “Transcriptional effects of gene CNAs were identified in 13 of these 43 robust TCs, of which four had transcriptional effects of CNAs at chromosome 17q.” Results line 105: “Fig. 2 summarizes these findings” I'm unsure where and how Figure 2 demonstrates this; Can you more clearly articulate and signpost to where

these results are in the Figures, by detailing each and every panel individually as the results are being described?

We completely agree with the reviewer that the previous version of the manuscript may have lacked detail in this section. We have expanded the section to provide more thorough explanations and context. The revised section now reads:

“Data regarding DFS were retrieved for 806 patients with colon cancer and 30 patients with rectal cancer. Patient characteristics and clinicopathological information are summarized in Table 1. We identified 53 DFS-associated primary TCs, of which 43 were robust. Fig. 2 provides an overview of the biological processes enriched in the DFS-associated primary TCs. The TCs are ordered based on their association with DFS, represented as $-\log^{10}(\text{p-value})$. This figure indicates robust TCs with solid circles, while non-robust TCs are indicated with open circles. The left-hand side of Fig. 2 displays the results of GSEA focusing on Hallmark gene sets. What stands out is that many of the TCs most strongly associated with DFS were related to epithelial-mesenchymal transition (EMT). The chromosomal location of given CNAs, for which the TCs capture downstream effects on gene expression levels, is also presented in the middle. Transcriptional effects of gene CNAs were identified in 13 of these 43 robust TCs, four of which were specifically linked to chromosome 17q. Among these four TCs, TC148, TC149, and TC189 ranked fifth, seventh, and eighth in terms of their significance in association with DFS. On the right-hand side of Fig. 2, two heatmaps detail the direction and association between these DFS-associated TCs and various clinicopathological parameters. Notably, higher activity in many of the strongest DFS-associated TCs correlated with increased inferred abundance of specific cell types in the tumor microenvironment, particularly fibroblasts, endothelial cells, neutrophils, myeloid dendritic cells, monocytic lineage cells, and dendritic cells.” Page 11, lines 7-23 and page 12, lines 1-2.

Remark 6.

Results line 90-95: “These primary TCs were found to be enriched for at least one gene set as defined in the 13 collections used. For example, 28% (61/220) of the primary TCs were enriched for at least one gene set from the Hallmark collection. The median top enrichment score (i.e., z transformed p-value) for the Hallmark gene set collection for all 61 primary TCs was 7.24(range: 4.07–28.06, IQR: 5.45–12.44). In addition, the transcriptional effects of copy number alterations (CNAs) were detected in 93 (42%) of the 220 primary TCs.” I was quite lost as to where each of these individual and specific points relate to in the figures provided, and quite how they were arrived at. I have no doubt that this method is drawing out new and important information, but given the amount of information presented in Figure 1 alone, there needs to be far more detailed alignment between the text and the figures. Currently the structure of the paper means that this section and its relationship with Figure 1 is hard to interpret.

We believe that the aforementioned restructuring of the manuscript has enhanced its clarity. Please see the response to Reviewer 2 Remark 2 for a more extensive explanation.

Remark 7.

Line 111-112 – “we performed random survival forest (RSF) analyses” – what is RSF and why was it chosen over other more standard methods?

We have incorporated the following explanation into the results section to elucidate our choice of statistical methods:

“RSF is a specialized statistical approach tailored for survival analysis. It is specifically advantageous for handling censored survival data while also offering predictive capabilities for time-to-event outcomes. The reasons for opting for RSF over alternative survival analysis methods lie in its inherent ability to manage complex datasets characterized by non-proportional hazards and nonlinear predictor-survival outcome relationships.” Page 12, lines 15-19.

Remark 8.

Line 114 – “We found that TC21 and TC117 had the highest importance scores” – what are the importance scores, how are they derived?

We had initially included the following description in the methods section: “Thus, for each RSF we calculated an ‘importance score’ for each TC; this score reflects how often that TC is an important classifier in the 1000 survival trees in the forest.” To enhance clarity, we have also included this explanation in the results section.

Overall, I really do value this story and hope that some additional detail to guide the reader through the process and results will help it to become a more utilised methods in future studies.

Reviewers' comments:

Reviewer #1 (Remarks to the Author):

The manuscript has undergone significant improvement, and the authors have addressed the majority of my queries. However, a few comments are noted:

1. Regarding Supplementary Figure 4, the inclusion of metadata information, such as MSI, CMS, iCMS, etc, as a colorbar on the side of the heatmap would be beneficial. This addition would facilitate a comprehensive understanding of how these 9 TCs stratify the patients.
2. The term "mean mixing matrix weights" requires further explanation. It is recommended to include a brief explanation or an equation in the Methods section that explains this concept.
3. Comments regarding "response to remarks 5 and 6":
 - a. Initially, my request was for a heatmap analogous to Supplementary Figure 4, with the additional of clinicopathological information. For this heatmap, each gene can actually be weighted to still capture importance genes that are subtle yet highly relevant for DFS in CRC. This kind of heatmap can help reader assessing the goodness of TCs that are discussed in the manuscript.
 - b. The authors brought attention to a noteworthy limitation associated with bulk transcriptomics data. I contend that addressing this limitation can be achieved through the utilization of single-cell data. For example, they have shown in figure 4 that TC202 and TC116 are more likely to expressed in stroma compartments. The authors now can check the expression of genes in this TC on stroma's cell type. Do they find any specific or new cell type expressing this TC?
 - c. Lastly, it is not clear which data they used for figure 4. The link they provided (<https://www.10xgenomics.com/resources/datasets>) contains many data from several different technology.
4. Similar to point 1, adding clinicopathological information in supplementary figure 3 will be helpful. The color scheme also very hard to be distinguished and should be improved.
5. The authors write "In total nine DFS-associated TCs were identified that are associated with EMT (Fig. 2)" but there is no explicit writing in the manuscripts, which TCs are they. Only 4 related to CMS4 are mentioned, but the other 5 never explicitly mentioned. The reader can be help with more clarity here.

Reviewer #2 (Remarks to the Author):

The authors have clearly addressed the main points around requests for more information and more detailed assessments, and I thank them for these additions. I have no further major comments that require new data, and congratulate the authors on their work.

In Remark 6, I had indicated that i felt that the text in the manuscript didn't fully explain all the information in the Figures; using an example of Figure 2 which the authors have now expanded in the text. However I also used the example of Figure 1 which has multiple panels but is only referred to once in the manuscript (even in the revised version) which makes it difficult for me as a reader to piece together exactly what each of the panels are showing relative to the results throughout.

For ease of reading, or indeed to clearly signpost the reader to which aspects of the overall study described in Figure 1 relate to data resented in the study, can the authors make a further attempt to slowly describe this initial figure using separate references to Fig 1A, 1B, 1C, 1D, 1E, 1F, 1G, 1H.. and

throughout the rest of the manuscript where multi panels have been used.

Letter addressing changes made in COMMSMED-23-0299-T entitled "Independent transcriptional patterns reveal biological processes associated with disease-free survival in early colorectal cancer"

We would like to thank the editor for considering our manuscript, COMMSMED-23-0299-T, entitled 'Independent transcriptional patterns reveal biological processes associated with disease-free survival in early colorectal cancer' for publication in Communications Medicine and for the chance to revise and resubmit the manuscript. We also want to thank both reviewers. Their previous comments and insights have greatly improved our manuscript. The new comments further help the clarity of our manuscript.

Responses to remarks from reviewers

Reviewer #1

The manuscript has undergone significant improvement, and the authors have addressed the majority of my queries. However, a few comments are noted:

Remark 1: Regarding Supplementary Figure 4, the inclusion of metadata information, such as MSI, CMS, iCMS, etc, as a colorbar on the side of the heatmap would be beneficial. This addition would facilitate a comprehensive understanding of how these 9 TCs stratify the patients.

We concur with the reviewer's suggestion to include metadata information in Supplementary Figure 4. We have now included TNM Stage, BRAF status, KRAS status, CMS classification, and MSI/MSS.

Remark 2: The term "mean mixing matrix weights" requires further explanation. It is recommended to include a brief explanation or an equation in the Methods section that explains this concept.

We agree with the reviewer. To explain the term 'mean mixing matrix weights' better in the main manuscript, the following has been added in the methods section: "In short, in the dataset containing mRNA expression profiles of p genes from n samples, the output of an ICA includes two matrices: (i) an independent component matrix with dimensions $i \times p$, where each weight within the independent components represents both the direction and magnitude of its effect on the expression levels of each gene, and (ii) a mixing matrix with dimensions $i \times n$, which contains the coefficients. These coefficients are indicative of the activity levels of each independent component within individual samples. Principal component analysis was performed on the covariance matrix between samples, after which the minimum number (representing i above) of top principal components that captured at least 85% of the total variance in the dataset was selected. The original mRNA expression level can be reconstructed by taking the inner product of the mixing matrix coefficients and the weights of the independent components for each gene. In ICA, an initial random weight vector with a variance of one must be chosen to achieve statistically independent components. Consequently, varying the initial random weight vectors can lead to different sets of independent components. To obtain a consensus set of independent components (referred to as

TCs), 25 ICA runs were conducted, each with a unique random initialization weight vector. Upon completion of all ICA runs, independent components with an absolute Pearson correlation coefficient greater than 0.9 were clustered, ensuring that the number of independent components in each cluster did not exceed the total number of ICA runs. TCs were derived by averaging the independent components within each cluster. Next, a credibility index for each TC was calculated by dividing the number of independent components in its cluster by the total number of ICA runs (25 in this case). TCs with a credibility index of 50% or higher were selected for constructing the TC matrix and the consensus mixing matrix.” Page 6, lines 5-22 and page 7, lines 1-3.

Remark 3A: Comments regarding "response to remarks 5 and 6": Initially, my request was for a heatmap analogous to Supplementary Figure 4, with the additional of clinicopathological information. For this heatmap, each gene can actually be weighted to still capture importance genes that are subtle yet highly relevant for DFS in CRC. This kind of heatmap can help reader assessing the goodness of TCs that are discussed in the manuscript.

We have created the suggested heatmaps for TC208 and TC117. These TCs were selected because TC208 showed the strongest association with DFS and the highest importance score in the random survival forest analysis for stage 3 colon cancer. Similarly, TC117 showed the second-highest association with DFS in patients with CRC and obtained the highest importance score for stage 2 colon cancer. The heatmaps are presented on page 3 and 4 of this letter. However, we have decided not to include these figures in the final manuscript as the complex interplay of numerous genes within each TC, as revealed by c-ICA, provides a more comprehensive understanding than the analysis of individual top genes alone.

Heatmap outlier genes TC208.

Heatmap outlier genes TC208.

Remark 3B: The authors brought attention to a noteworthy limitation associated with bulk transcriptomics data. I contend that addressing this limitation can be achieved through the utilization of single-cell data. For example, they have shown in figure 4 that TC202 and TC116 are more likely to be expressed in stroma compartments. The authors now can check the expression of genes in this TC on stroma's cell type. Do they find any specific or new cell type expressing this TC?

We agree with the reviewer's suggestion and acknowledge that integrating these methodologies will enhance our understanding of the biology of early colon cancer. To this end, we utilized single-cell data from the Gut Cell Atlas (<https://www.gutcellatlas.org/#datasets>). The procedure adopted is described in the methods section, which states: "Single-cell data in h5ad format were obtained from the Gut Cell Atlas (<https://www.gutcellatlas.org/#datasets>), encompassing a comprehensive single-cell RNA-seq dataset of 428,000 intestinal cells from fetal, pediatric, and adult donors across up to 11 distinct intestinal regions. Our analysis focused specifically on the mesenchyme lineage. We downloaded the normalized data and associated metadata for this lineage. We then implemented a sub-sampling strategy, randomly selecting 10% of cells from the mesenchyme lineage dataset, resulting in an analysis cohort of approximately 16,000 cells. For the projection analysis, we employed 3,000 permutations with the Johnson transformation. To facilitate the visualization and interpretation of the corrected mixing matrix weights, box and whisker plots were created for each cell type within every TC panel." Page 9, lines 20-24 and page 10, lines 1-5.

Furthermore, the following paragraph was added to the results section: "To gain insight into the role of specific mesenchymal cells on the biology captured by the EMT-related, DFS-associated TCs, we utilized single-cell transcriptome data obtained from the Gut Cell Atlas (<https://www.gutcellatlas.org/#datasets>). This dataset is a comprehensive collection of single-cell RNA-seq profiles of 428,000 intestinal cells from fetal, pediatric, and adult donors, across up to 11 distinct intestinal regions. Supplementary Figure S6 features box and whisker plots that illustrate the activity of the EMT-related DFS-associated TCs in various mesenchymal cell types. These plots provide valuable insights into the specific mesenchymal cell populations that might influence the biological processes associated with DFS. For example, mesothelium cells show pronounced higher activity of TC208, while pericytes show higher activity of TC117. Additionally, our analysis showed higher activity of TCs 202 and 136 in the stromal 4 cell type — a newly identified stromal population enriched for pro-inflammatory and fibroblastic reticular cell genes.²²" Page 16, lines 14-23 and page 17, lines 1-2.

Remark 3C.

Lastly, it is not clear which data they used for figure 4. The link they provided (<https://www.10xgenomics.com/resources/datasets>) contains many data from several different technology.

We now specified the spatial transcriptomic profiles were retrieved from the 10x website <https://www.10xgenomics.com>, in the spatial gene expression dataset.

Remark 4.

Similar to point 1, adding clinicopathological information in supplementary figure 3 will be helpful. The color scheme also very hard to be distinguished and should be improved.

We concur with the reviewer's suggestion to include metadata information in Supplementary Figure 3. We have now included TNM Stage, BRAF status, KRAS status, CMS classification and MSI/MSS. We have also upgraded the color scheme.

Supplementary Figure S3

Remark 5.

The authors write "In total nine DFS-associated TCs were identified that are associated with EMT (Fig. 2)" but there is no explicit writing in the manuscripts, which TCs are they. Only 4 related to CMS4 are mentioned, but the other 5 never explicitly mentioned. The reader can be help with more clarity here.

We agree with the reviewer. The nine EMT-associated TCs were previously not specifically mentioned. This has now been updated, the text now reads: "What stands out is that many of the TCs most strongly associated with DFS were related to epithelial-mesenchymal transition (EMT). In particular, the following nine TCs demonstrated a strong association: TC208, TC117, TC193, TC202, TC116, TC153, TC58, TC136, and TC77." Page 13, lines 2-4.

Reviewer #2

The authors have clearly addressed the main points around requests for more information and more detailed assessments, and I thank them for these additions. I have no further major comments that require new data, and congratulate the authors on their work.

Remark 1.

In Remark 6, I had indicated that i felt that the text in the manuscript didn't fully explain all the information in the Figures; using an example of Figure 2 which the authors have now expanded in the text. However, I also used the example of Figure 1 which has multiple panels but is only referred to once in the manuscript (even in the revised version) which makes it difficult for me as a reader to piece together exactly what each of the panels are showing relative to the results throughout.

For ease of reading, or indeed to clearly signpost the reader to which aspects of the overall study described in Figure 1 relate to data resented in the study, can the authors make a further attempt to slowly describe this initial figure using separate references to Fig 1A, 1B, 1C, 1D, 1E, 1F, 1G, 1H, and throughout the rest of the manuscript where multi panels have been used.

We agree with the reviewer's suggestion and have enhanced the manuscript's readability by explicitly referencing the figure panels within the main text where they are discussed in detail.

REVIEWERS' COMMENTS:

Reviewer #1 (Remarks to the Author):

The authors have addressed all my concerns